# Engineering grain boundaries at the 2D limit for the hydrogen evolution reaction

Yongmin He[1,2,20], Pengyi Tang [3,4,20], Zhili Hu[5,6,20], Qiyuan He [1,20], Chao Zhu[1], Luqing Wang[6], Qingsheng Zeng[1], Prafful Golani[1], Guanhui Gao[7], Wei Fu[1], Zhiqi Huang[1], Caitian Gao[8], Juan Xia[9], Xingli Wang[10], Xuewen Wang[11], Chao Zhu [1], Quentin M. Ramasse[12,13], Ao Zhang [1], Boxing An[14], Yongzhe Zhang[14], Sara Martí-Sánchez[3], Joan Ramon Morante [4], Liang Wang [15], Beng Kang Tay [10], Boris I. Yakobson[6], Achim Trampert[7], Hua Zhang [1,16], Minghong Wu [15]*, Qi Jie Wang [2,17]*, Jordi Arbiol [3,18]* & Zheng Liu [1,8,17,19]*

Atom-thin transition metal dichalcogenides (TMDs) have emerged as fascinating materials and key structures for electrocatalysis. So far, their edges, dopant heteroatoms and defects have been intensively explored as active sites for the hydrogen evolution reaction (HER) to split water. However, grain boundaries (GBs), a key type of defects in TMDs, have been overlooked due to their low density and large structural variations. Here, we demonstrate the synthesis of wafer-size atom-thin TMD films with an ultra-high-density of GBs, up to ~$10^{12}$ $cm^{-2}$. We propose a climb and drive 0D/2D interaction to explain the underlying growth mechanism. The electrocatalytic activity of the nanograin film is comprehensively examined by micro-electrochemical measurements, showing an excellent hydrogen-evolution performance (onset potential: −25 mV and Tafel slope: 54 mV $dec^{-1}$), thus indicating an intrinsically high activation of the TMD GBs.

[1] School of Materials Science and Engineering, Nanyang Technological University, Singapore 639798, Singapore. [2] Center for OptoElectronics and Biophotonics, School of Electrical and Electronic Engineering & The Photonics Institute, Nanyang Technological University, Singapore 639798, Singapore. [3] Catalan Institute of Nanoscience and Nanotechnology (ICN2), CSIC and BIST, Campus UAB, Bellaterra, Barcelona 08193 Catalonia, Spain. [4] Catalonia Institute for Energy Research (IREC), Jardins de les Dones de Negre 1, Sant Adrià del Besòs, Barcelona 08930 Catalonia, Spain. [5] College of Aerospace Engineering, Nanjing University of Aeronautics and Astronautics, Nanjing 210016, China. [6] Department of Materials Science and NanoEngineering, Rice University, Houston, TX 77005, USA. [7] Paul-Drude-Institut für Festkörperelektronik Leibniz-Institut im Forschungsverbund Berlin Hausvogteiplatz, 5-7, 10117 Berlin, Germany. [8] Centre for Micro-/Nano-electronics (NOVITAS), School of Electrical & Electronic Engineering, Nanyang Technological University, 50 Nanyang Avenue, Singapore 639798, Singapore. [9] Institute of Fundamental and Frontier Sciences, University of Electronic Science and Technology of China, Chengdu 610054, China. [10] CNRS-International-NTU-THALES Research Alliance, Nanyang Technological University, Singaproe 637553, Singapore. [11] Institute of Flexible Electronics, Northwestern Polytechnical University, Xi'an 710072, China. [12] SuperSTEM Laboratory, SciTech Daresbury Campus, Keckwick Lane, Daresbury WA44AD, UK. [13] School of Chemical and Process Engineering, University of Leeds, Leeds LS29JT, UK. [14] College of Materials Science and Engineering, Beijing University of Technology, Beijing 100124, People's Republic of China. [15] School of Environmental and Chemical Engineering, Shanghai University, Shanghai 200444, China. [16] Department of Chemistry, City University of Hong Kong, Tat Chee Avenue, Kowloon, Hong Kong, China. [17] CINTRA CNRS/NTU/THALES, UMI 3288, Research Techno Plaza, Singapore 637553, Singapore. [18] ICREA, Pg. Lluís Companys 23, Barcelona 08010 Catalonia, Spain. [19] Environmental Chemistry and Materials Centre, Nanyang Environment and Water Research Institute, Singapore, Singapore. [20]These authors contributed equally: Yongmin He, Pengyi Tang, Zhili Hu, Qiyuan He. *email: mhwu@shu.edu.cn; qjwang@ntu.edu.sg; arbiol@icrea.cat; z.liu@ntu.edu.sg

Grain boundaries (GBs) are commonly found in atom-thin or so called two-dimensional (2D) polycrystalline materials[1–3], where they can be described as line defects. They play a key role in shaping the properties and performance of 2D materials in applications as varied as mechanical strengthening[4], photovoltaics[5,6], electronics[7–9], and catalysis[10,11]. Engineering the structure and/or the density of GBs in 2D materials could thus become a promising way to tailor their performance. One particular class of 2D materials, transitional metal dichalcogenides (TMDs) have attracted a great deal of attention for its possible uses in electrocatalytic reactions[12,13], including the hydrogen evolution reaction (HER)[14,15]. Due to the low cost, earth abundance and good stability of a wide range of TMDs, substantial work has thus been undertaken to improve their electrocatalytic activity[12,13] by, e.g., exposing their edges[16–19], doping with heteroatoms[20], and/or creating and straining structural defects[11,15,19,21,22]. In contrast, less attention has been paid to the role of GBs due to their typically low number density and large structural variations, even though GBs have been predicted to be highly electrocatalytically active[23]. The poor control over the density and structure of GBs stems from the fast gaseous kinetic processes and the multiplicity of chemical phases involved during the growth of TMDs[2,24,25]. To date, the most common methods employed to synthesize atomically thin polycrystalline TMDs include chemical vapor deposition (CVD)[2,3,26–36], physical vapor deposition (PVD)[37–39], and metal organic chemical vapor deposition (MOCVD)[40,41]. Films grown using these techniques usually exhibit grain sizes ranging from hundreds of nanometers to several millimeters, resulting in a low GB density, as summarized in Fig. 1a (See the statistical method in Supplementary Fig. 1).

Here, we fabricate wafer-size atomically thin TMD films with sub-10 nm grains by means of Au-quantum-dots (QDs)-assisted vapor-phase growth, demonstrating an ultra-high-density of GBs (up to ~$10^{12}$ cm$^{-2}$). The quality of the films was examined by high-resolution transmission electron microscopy (HRTEM), aberration-corrected scanning transmission electron microscopy (STEM), scanning electron microscopy (SEM), and Raman spectroscopy. Experimental evidence as well as phase-field simulations demonstrate that the Au QDs regulate the formation of the TMD grains. We then investigated the catalytic activity of these nanograin films by a four-electrode micro-electrochemical cell for hydrogen evolution. An excellent performance (−25 mV and 54 mV dec$^{-1}$ for the onset potential and the Tafel slope) was obtained for our MoS$_2$ nanograin films, indicating a good intrinsic activation of the GB-rich 2D basal plane.

## Results

### Controlled growth of the TMD nanograin film.
The main obstacles that hinder the vapor growth of TMDs at the 2D limit to grain sizes typically <10 nm consist of the precise control of the nucleation sites and the growing rate of the grains. These can be addressed by using a high-density of Au QD seeds and a low mass flow rate of the vapor sources. Figure 1b illustrates the wafer-scale growth process (see also a full description of the growth setup in Supplementary Fig. 2). A wafer-scale Au QD layer was fabricated on 2-inch sapphire or SiO$_2$/Si substrates and used to subsequently grow the atom-thin MoS$_2$ films. Figure 1c shows the geometries of the as-grown wafer-size film from the centimeter to the nanometer scale. The SEM image (Fig. 1d) shows the as-prepared Au QDs with an ultra-high density up to ~$2 \times 10^{12}$ cm$^{-2}$ and average diameter down to ~4.8 nm, which was achieved by heating a thin Au film on sapphire or SiO$_2$/Si substrates at a high temperature. We attribute the formation process of these Au QDs to a solid-state dewetting behavior at the interface between the SiO$_2$ (or

Al$_2$O$_3$) and Au[42,43] (Supplementary Fig. 3). The initial deposition time of the Au films prior to heating determines the density and the size of the final Au QD structures: a shorter deposition time results generally in a smaller size and a higher density of Au QDs, as shown in Fig. 1e and Supplementary Fig. 4. The MoS$_2$ film is then grown using a vapor-phase growth technique (see further experimental details in Method section), after which Au QDs can be removed from as-grown films, by using KI/I$_2$ etchant at room temperature. The successful removal of the Au QDs after the etching treatment was assessed by X-ray photoelectron spectroscopy and STEM imaging in Supplementary Figs. 5–7, showing only minimal amounts of residual Au present. The resulting MoS$_2$ films were examined by Raman spectroscopy, as shown in Fig. 1f. The Raman spectroscopy measurements indicate that the MoS$_2$ film is 1–3 layers thick (1–3 L). A weak in-plane mode (E$_{2g}^1$ at 385 cm$^{-1}$) was observed in our 1–3 L film, with an intensity ratio with the out-of-plane mode A$_{1g}$ (at 405 cm$^{-1}$), $I_{E_{2g}^1}/I_{A_{1g}}$ (~0.28), significantly smaller than the ratios typically measured on the CVD-grown (~1.89) and exfoliated (~1.57) MoS$_2$ films in our experiment, suggesting a highly polycrystalline structure in this film (see the details in Supplementary Table 1 and Supplementary Note 1).

Au QDs regulate the formation of TMD grains in two ways. The first way is that Au QDs can facilitate the TMD nucleation at the initial stage of the TMD growth. The melting temperature of Au will dramatically decrease as its size decreases to several nanometers, as shown in Supplementary Fig. 8. As a result, at the growth temperature of TMDs (650–800 °C), Au QDs tend to be in the liquid phase due to their small size. Such liquid Au droplets can facilitate the formation of the TMD nucleation sites during the growth process. The second way is that the Au QDs can confine the size of the TMD domains below 10 nm. Our deposition method can produce well-dispersed Au QDs with a very high number density, with average spacing between QDs down to a few nanometers. As a result, the formation of the TMD grains will be constrained at this length scale.

### Atomic structure of the TMD nanograin films.
We investigated the atomic structure of the TMD nanograin films using transmission electron microscopy (TEM). Figure 2a shows an overview TEM image of a uniform and continuous MoS$_2$ film suspended on a Cu-supported lacey carbon TEM grid. This region is representative of the tens of samples examined, and the MoS$_2$ film comprises polycrystalline patches with regions of 1–3 L MoS$_2$ (consistent with the measured Raman frequencies[44]). To evaluate the grain distribution, we randomly chose six regions of the film, labeled 1–6 on Fig. 2a, for further HRTEM imaging. The original HRTEM images from these six regions, corresponding Fourier transforms (FFTs) and frequency filtered images (IFFTs)[45] are shown in the top (a1–a6), middle (b1–b6), and bottom panels (c1–c6) of Fig. 2a, respectively. To aid the visualization of the different grains, black-dashed lines are added along the edges of the grains in the bottom panel (c1–c6). From these images, we can identify 8–10 distinct MoS$_2$ grains in a ~700 nm$^2$ region, suggesting an ultra-high grain density (up to ~$10^{12}$ cm$^{-2}$ in this representative patch, consistent with the initial estimate derived from the Au QD number density) in our sample. Accordingly, the average diameter of the grains is <10 nm, with some observed grains smaller than 5 nm in diameter. To our knowledge, this is the smallest grain size obtained to date in materials at the 2D limit[17,46,47].

We then employed atomic-resolution high-angle annular dark-field (HAADF) imaging in the STEM to examine the atomic structure of the GBs on the as-grown nanograin films, representative examples of which are shown in Fig. 2b, c and

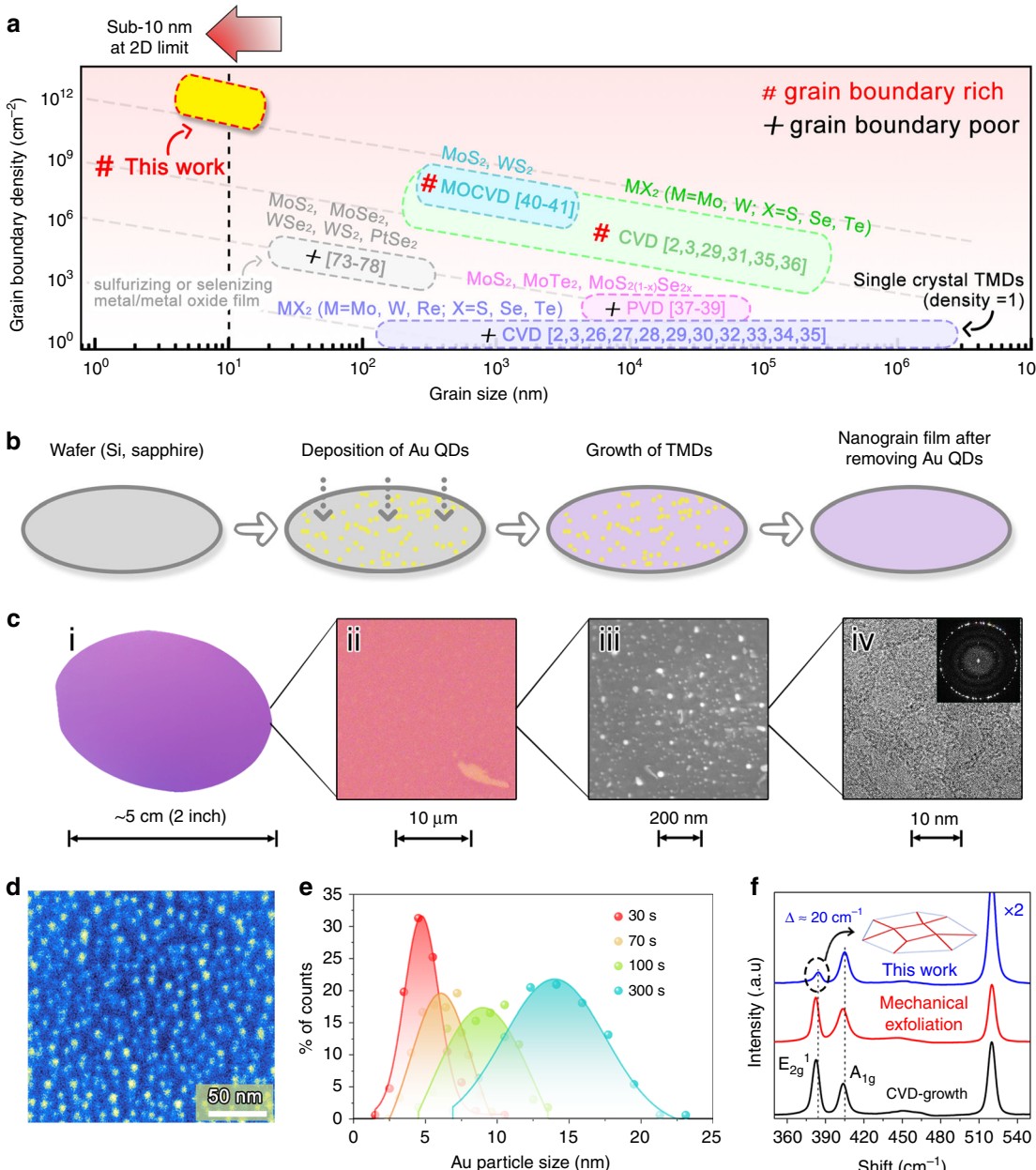

**Fig. 1 Synthesis of wafer-size TMD nanograin films. a** Overview of the grain size and density in TMD materials obtained by various fabrication methods, such as CVD[2,3,26-36], MOCVD[40,41], PVD[37-39], top–down syntheses of sulfurization/selenization of metal and metal oxide thin films[73-77], and thermal decomposition of thiosalts thin films[78]. Although small grains (~20 nm in diameter) were observed in TMDs synthesized by the top–down methods of sulfurizing Mo-based thin films[73,74], or thermal-decomposition of Mo-thiosalts thin films[78], the grain density in the film is not very high owing to their interlayer diffusion synthesis mechanism. **b** Schematic of the wafer-scale growth of TMD nanograin films. Ultra-high-density Au quantum dots (QDs) were used to grow the $MoS_2$ (as well as $WS_2$ atom-thin films: see Supplementary Information). **c** TMD nanograin film from wafer scale to nanoscale, including photograph (i), optical image (ii), SEM image (iii), and HRTEM image (iv). **d** False-colored SEM image of the Au QDs on a $SiO_2$/Si substrate, showing an average diameter of 4.8 nm. **e** Statistical distributions of Au QDs obtained from Au films with different deposition times. The evaporation rate is ~0.1 Å s$^{-1}$ in our experiments. **f** Raman spectra acquired from the $MoS_2$ film. The difference ($\Delta$ ~20 cm$^{-1}$) between the out-of-plane ($A_{1g}$) and in-plane ($E^1_{2g}$) Raman modes indicates that the $MoS_2$ film is 1–3 layers (1–3 L)[2,44]. Compared to CVD-grown and exfoliated samples, the reduced intensity of the in-plane mode indicates a highly polycrystalline structure for this $MoS_2$ film.

Supplementary Fig. 9. The inspection of dozens of boundary locations systematically found that nanograins in our films are stitched by GBs. The atomic structure of the GBs varied significantly (great care was taken to exclude any structure modification arising from beam damage, see Experimental section for details). Structures comprising a combination of 5|7 and 6|8 rings were however dominant in our nanograin films and are illustrated in Fig. 2c. A small number of 8|4|4 structures were

found, but in contrast, 12|4 rings were never observed in "pristine" GBs (i.e., before any substantial exposure to the electron beam). Their atomic structures in $MoS_2$ are schematically illustrated in Supplementary Fig. 10.

Although the nature of atomic-resolution microscopy precludes large-scale statistical studies, we investigated as many different regions in a $MoS_2$ nanograin film as practically possible. Six further regions over a large range were carefully examined

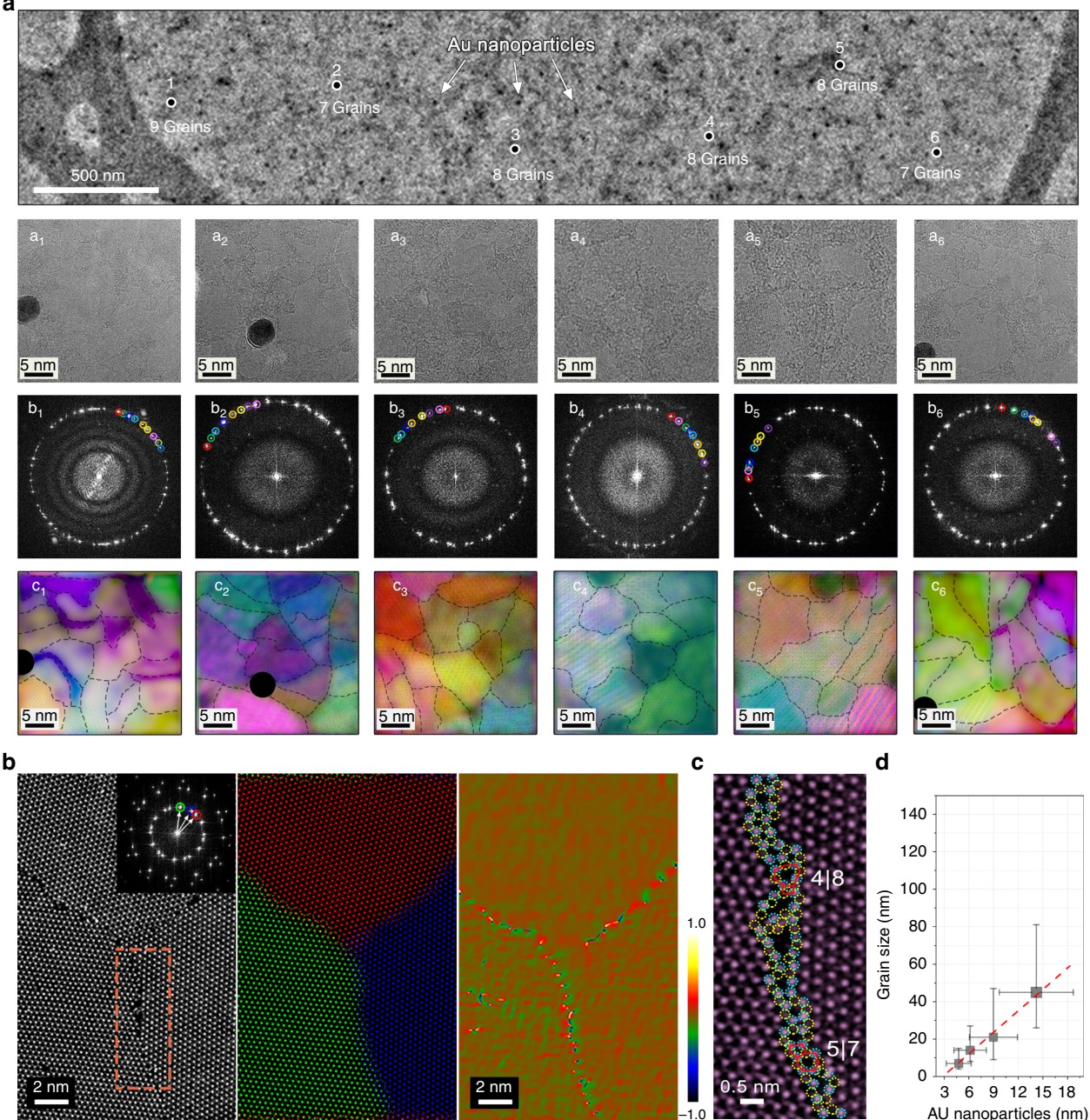

**Fig. 2 Atomic structure of the TMD nanograin film. a** TEM image of the 1–3 L MoS$_2$ nanograin film, showing a uniform and continuous atom-thin layer with Au nanoparticles (dark spots). Top panel (a1–a6): HRTEM images from regions 1–6. Middle panel (b1–b6): Fourier transforms (FFTs) of regions 1–6 from the corresponding TEM image. Bottom panel (c1–c6): false-colored frequency filtered (IFFTs) images of the same regions. Grains with different orientations give rise to distinct sets of rotated spot patterns in the FFTs, as indicated by colored circles in the middle panel (b1–b6); in turn, the localization of the grains can be visualized in real space by color coding the corresponding contribution to the IFFT of a given set of diffraction spots, and overlaying the IFFTs into a composite colored image (in bilayer regions, the colors are combined). In order to better visualize the different grains, the grains are highlighted in the IFFT images with black-dashed lines. **b** HAADF STEM investigation of MoS$_2$ grains. Left: HAADF STEM image showing the GBs between three MoS$_2$ grains. Inset on the left figure: Fourier transform of the image showing three distinct sets of MoS$_2$ diffraction patterns with rotation angles of 11.6° and 25.6°. Middle: composite color-coded IFFT image. Right: dilatation map of **b** after applying GPA routine to the monolayer MoS$_2$. **c** Higher magnification atomic resolution HAADF STEM image of the brown-dashed line marked region in **b**, showing the detailed atomic structure of the GB, which is composed of 5|7 and 4|8 rings. Mo atoms are marked with indigo dotted circles and S atoms marked with yellow dotted circles. **d** Grain size dependence on Au nanoparticle size (average diameter) in the MoS$_2$ nanograin film. The error bars of Au nanoparticles are extracted from the full-width at half maximum in Fig. 1e.

with atomically resolved HAADF–STEM imaging (Supplementary Fig. 11), and 3 to 5 GBs can be observed in a 400 nm$^2$ area, confirming the ultra-high-density of GBs observed in STEM (consistent with the ~10$^{12}$ cm$^{-2}$ suggested by the Au QD density). Moreover, tens of samples with different sizes of Au QDs were grown: the grain size measured in the resulting MoS$_2$ films exhibits a clear linear relationship with the QD size, as shown in Fig. 2d. This provides a strong proof that the TMD

grain size can be carefully controlled by the Au QD substrate. Remarkably, our method can be also extended to other TMD materials such as WS$_2$ (Supplementary Fig. 12 summarizes similar results to those described above, using WS$_2$ instead of MoS$_2$), making it as a universal approach for the wafer-size synthesis of atom-thin films with sub-10 nm grains.

**Growth mechanism of the TMD nanograin film.** A climb and drive zero-dimensional (0D)/2D interaction is proposed to explain the Au QDs-assisted-growth mechanism of the TMD nanograins at the atomic limit, which is supported by real-time phase-field simulations and experimental evidence. The phase-field simulation of the climb process shows that, once the Au QDs encounter a MoS$_2$ edge at the growth front, they migrate swiftly from the SiO$_2$ surface onto the MoS$_2$ surface (Fig. 3a). This behavior is mainly attributed to a smaller wetting angle of the Au QDs droplet (Au is in the liquid phase at the growth temperature) on MoS$_2$ than that on the SiO$_2$ substrate[48] (Supplementary Fig. 13). This phenomenon is also experimentally confirmed by cross-sectional TEM, as shown in Supplementary Fig. 14, where nearly all the Au QDs sit on the surface of the MoS$_2$. Subsequently, the phase-field simulation suggests the formation of a second MoS$_2$ layer will drive the Au QD droplets along its growth direction (Fig. 3b). A number of the Au QDs may thus coalesce and form bigger droplets during the driving process. This is supported by our experimental evidence: the size of the Au QDs on the MoS$_2$ is usually larger than those on SiO$_2$, and some of them sit at the edge of the MoS$_2$ sheet, as illustrated in Fig. 3c and Supplementary Fig. 15a–b. This drive process will be immobilized when the Au QDs reach a critical size (as a result of the Au returning to the solid phase). This would be consistent with experimental observations that larger Au particles seen in the minority multilayer regions of the nanograin film appear to be encapsulated by the MoS$_2$ film, suggesting they were immobilized after reaching a critical size, leading the MoS$_2$ to grow over them rather than driving them forward (see Supplementary Figs. 15c, d and 16).

Figure 3d shows successive snapshots that illustrate the growth of TMD nanograins films via a phase-field simulation (also see Supplementary Movie 1). The Au QDs serve as nucleation sites to form the first MoS$_2$ layer (Fig. 3d–i). The growth of a second MoS$_2$ layer will then drive the Au QDs. Some of the Au QDs will be immobilized at the GBs (Fig. 3d ii and iii, number 1–4). More interestingly, we found that neighboring grains could be sutured together through a zipper effect (Fig. 3d iii-v, number 5 and 6, see Method and Supplementary Fig. 17). It is important to emphasize that the growth model for the TMD nanograin films we describe here is driven by a 0D/2D interaction at the atomic limit, which is distinct from the conventional metal-film-assisted growth of 2D materials (2D/2D)[29,35], or from the metal-nanoparticle-assisted growth of nanowires (0D/1D)[49].

**Hydrogen production of the TMD nanograin films.** We first performed first-principle calculations[50] to examine all of the HER catalytic active sites in model MoS$_2$ geometries (Fig. 4a), including the basal plane sites, edges with different passivation (Mo or S)[51], and GBs with different atomic structural configurations (5|7, 6|8, 4|6, 12|4, and 8|4|4 rings). The calculated hydrogen adsorption free energies ($\Delta G_H$)[52] of various atomic structures (Supplementary Fig. 18) are shown in Fig. 4b and Supplementary Table 2. The MoS$_2$ basal plane has a $\Delta G_H$ as high as 1.79 eV[15], indicating a HER-inert surface. For edges and GBs, we compare the activities of most of the experimentally observed structures, e.g., 5|7 (0.132 eV), 6|8 (−0.237 eV), and 8|4|4 (0.52 eV, −0.044 for defect) ring combinations,[2,53] 50% S passivated

Mo edge (0.561 eV), and 50% passivated S edge (0.446 eV)[53,54]. It can be seen that GBs indeed show comparable or even better activity than the edges in general, suggesting GBs are promising candidates as high-efficient catalyst sites. In addition, we note that some edges are not further considered here owing to their instability in air; for instance, the otherwise energetically favorable Mo edge without passivation of S atoms (−0.446 eV).

We then developed a micro-electrochemical cell to investigate the HER activity of our sub-10 nm nanograin films. Furthermore, the HER activity of model single-GBs (mirror GB)[3,11], single-edge (Supplementary Fig. 19)[53], and basal plane structures in MoS$_2$ samples from CVD method[1] were also examined for comparison. Figure 5a illustrates the micro-cell's structure. Figure 5b shows a picture of the micro-cell and electrodes configuration where a graphite counter electrode and a micro-reference electrode were used. In our experiment, a vertical MoS$_2$/graphene heterostructure was designed, in which graphene played two important roles: one is to efficiently inject electrons to MoS$_2$. Such a strategy has already been widely adopted in TMD-based semiconductor devices[55,56], while the other is to provide a fair comparison by eliminating the contact resistance variations of the individual MoS$_2$ electrodes onto the substrate. Previous work[56,57] suggested the formation of an inconsistent contact barrier between the MoS$_2$ and the Au electrode, owing to their complex metal–semiconductor interface leading to effects such as Fermi pinning, creation of alloy structures, etc. The formation of such a contact barrier was also studied by phase engineering[21] and field-effect gating[58] in the HER process. Considering the important role of this graphene supporting layer, we introduced one more working electrode in the micro-electrochemical cell (four-electrode micro-electrochemical cell[59,60], see circuit diagram in Supplementary Fig. 20) to monitor in situ its conductance. The device fabrication of the MoS$_2$/graphene heterostructure is detailed in Supplementary Figs. 21 and 22, and Supplementary Notes 2 and 3. The heterostructure was also characterized by Raman spectroscopy (Supplementary Fig. 23). Figure 5c–f shows the optical images obtained from the typical devices fabricated with a MoS$_2$ nanograin film, a single-GB model structure (the presence of a GB was confirmed by Raman mapping, as illustrated on Fig. 5d, right panel), a single edge, and a basal plane, respectively. In these optical images, only the exposed MoS$_2$ in the reaction window contributes to the electrocatalytic reaction, as indicated by the white arrows. The rest of the areas are covered by poly(methylmethacrylate) (PMMA), and the graphene supporting layer (Supplementary Fig. 24) are also electrochemically inert.

Figure 5g, h shows the polarization curves and the corresponding Tafel slopes in a 0.5 M H$_2$SO$_4$ solution and provide several interesting observations. First, the resistivity of the graphene supporting layer is found to be as low as $10^{-4}$ $\Omega$ mm during the HER (see the inset in Fig. 5g), indicating the excellent current injection performance of the graphene layer. Secondly, the single-GB device delivers a better activity than the single-edge device (see also the measurement of the active length per unit area of the GB in Supplementary Fig. 25), and both are superior to the basal plane device, echoing our theoretical calculation results shown above. More importantly, our MoS$_2$ nanograin film shows a remarkable HER performance: −25 mV and 54 mV dec$^{-1}$ for the onset potential and the Tafel slope, respectively.

In order to evaluate the HER performance of the TMD nanograin film accurately, we have fabricated hundreds of devices and tested them in a size-controlled micro-electrochemical cell. The HER data, including current density, Tafel slope, and onset potential of the nanograin film and other types of electrostatically active sites are presented in Fig. 6a–c and Supplementary Fig. 26. The window size is controlled from 25 to 150 μm$^2$. Our results

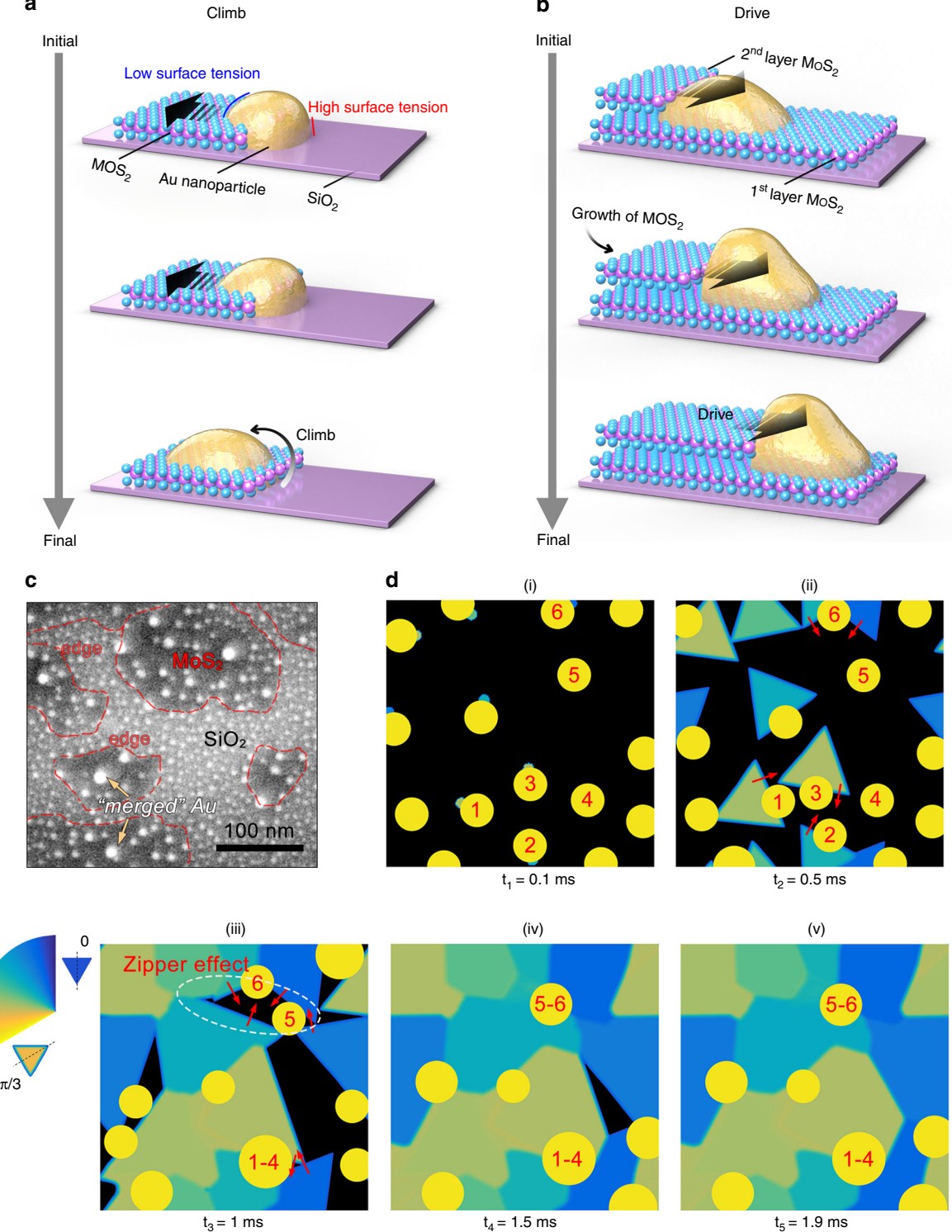

**Fig. 3 Proposed self-limiting climb and drive growth mechanism of the TMD nanograin films. a** Schematic of the climb stage: the Au QDs droplets tend to climb onto the $MoS_2$ monolayer from the $SiO_2$ substrate due to the surface tension difference. **b** Schematic of the drive stage: the growth of a second $MoS_2$ layer tends to push the Au QDs along the growth direction. Most of the Au QDs will be stabilized at the grain boundaries in order to minimize their surface energy. **c** SEM image showing the distribution of the Au QDs on the edges of growing $MoS_2$ layers (marked by the red-dashed line). Au QDs are usually larger on the $MoS_2$ layers (indicated by blue arrows) than those observed on the $SiO_2$ substrate. **d** Real-time phase-field simulation showing the formation of the $MoS_2$ nanograin film. The Au QDs numbered 1–4 show that the growth of the $MoS_2$ layer will push Au QDs toward the grain boundaries. Au QDs numbered 5–6 show a zipper effect to suture the neighboring grains together. The movie can be found in Supplementary Movie 1 (Real-time phase-field simulation of the growth process).

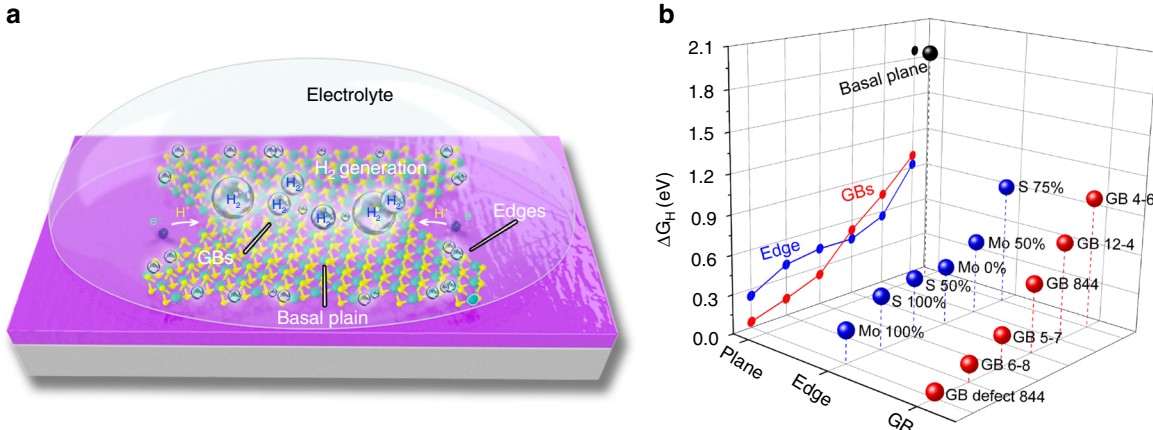

**Fig. 4 Theoretical calculations of the catalytic sites in MoS₂ for HER. a** Schematic of MoS₂ with an illustration of the main types of catalytically active sites for the hydrogen evolution reaction (HER) including: basal plane atomic sites, edges, and GBs. **b** corresponding $\Delta G_H$ of various types of catalytically active sites in MoS₂ catalysts. Some types of GBs (e.g., 844, 6–8, and 5–7) show better electrocatalytic activity than edges.

demonstrate the excellent HER performance of the nanograin film. The HER current density of the nanograin film is up to ~1000 mA cm⁻² while the Tafel slope and onset potential are down to ~50 mV dec⁻¹ and ~−25 mV, thus exhibiting a performance superior to devices using the basal plane of the CVD film, the single edge and the single GB. This suggests that sites on the otherwise HER-inert basal plane of the MoS₂ have been activated by the presence of GBs. The obtained HER performance is also comparable to the performance of strained sulfur vacancies[15] or the metastable 1T-phase[61], which are arguably more challenging to realize experimentally.

Finally, as a comparison, we have conducted conventional macro-cell measurements on a MoS₂ nanograin film on a glassy carbon electrode, as shown in Supplementary Figs. 27–32 and Supplementary Notes 4 and 5. A remarkable HER performance (−44 mV and 55 mV dec⁻¹ for the onset potential and the Tafel slope, respectively) of the MoS₂ nanograin film is observed (Supplementary Fig. 28), which is consistent with the results obtained in the micro-cell. It also shows an excellent long-term stability for HER, as shown in Supplementary Fig. 29. It is worth mentioning that Au single-atom exists in our MoS₂ nanograin film and may contribute to the overall HER performance. However, based on the structure (Supplementary Fig. 6) and extremely low content (Supplementary Fig. 7) of Au single atoms as well as their low HER-activity (Supplementary Figs. 33 and 34, and Supplementary Note 6), we conclude that the residual Au is not the main contributor of the good HER activity in MoS₂ nanograin film. The measured performance should be mainly from GB due to its superior activity and ultra-high density. As a proof-of-concept practical application, hydrogen production on a wafer-size (2 inch) MoS₂ nanograin film was also demonstrated (see Supplementary Movie 2). A large amount of H₂ bubbles could be produced during the reaction, demonstrating the tantalizing potential for practical application of GBs-engineered catalysts for hydrogen production.

## Discussion

In summary, we have engineered wafer-size ultra-high-density GBs at the 2D limit in TMD films. Phase-field simulations reveal a climb and drive 0D/2D-interaction-based growth mechanism. As a proof-of-concept application in electrocatalysis, devices based on our TMD nanograin films deliver a superior hydrogen evolution performance (onset potential: −25 mV and Tafel slope: 54 mV dec⁻¹), indicating an intrinsic activation of the GB-rich

2D plane. Beyond electrocatalysis, the nanograin films may provide further diverse potential applications, such as in resistance-memory devices, flexible devices, or for use as mechanical films and molecular sieving films.

## Methods

**Growth of water-size MoS₂ and WS₂ nanograin films.** The first step consists in depositing Au layers of various thicknesses on clean 2-inch sapphire or SiO₂/Si wafers by e-beam evaporation at the rate of 0.1 Å s⁻¹; these are used as the growth substrate for the nanograin films. Subsequently, Au-coated wafers are introduced into a CVD apparatus, which is depicted in detail in Supplementary Fig. 2. MoO₃ or WO₃ powders were loaded in an aluminum oxide boat and used as sources. Sulfur powder is placed in a second aluminum oxide boat upstream of the MoO₃ or WO₃ in the outer tube. In a third step, Ar (flow 100 sccm) and Ar/water vapor (flow 50 sccm) are introduced as carrier gases for the growth in the inner and outer tubes, respectively. The growth temperature was kept at 780 °C in low-vacuum conditions of 1–10 kPa; the S powder is separately kept at ~160 °C. Adjusting the growth time is used to control the thickness of films. For example, Condition (I): 3–5 min of the growth time for 1–3 layers (1–3L); Condition (II): 5–10 min of the growth time for 4–5 layers (4-5 L). The growth parameters for WS₂ nanograin films are similar to those used for MoS₂, except for a higher growth temperature (900 °C). Finally, Au nanoparticles can be completely removed from as-grown nanograin MoS₂ or WS₂ films by using a KI/I₂ etcher for 40–60 min at room temperature (see Supplementary Figs. 5 and 6).

Special care needs to be taken on two important points: (i) the Au QDs will form once the temperature is above 300 °C, due to the solid-state dewetting behavior[42,43,62] at the interface of SiO₂ (Al₂O₃) and Au (see Supplementary Fig. 3). Prior to the TMD growth, the substrate therefore stabilizes at this temperature to ensure the presence of Au QDs. (ii) Small amounts of water vapor are used to prevent the Mo or W sources from complete sulfurization before they reach the growth wafer, thus enabling a steady supply of sources during growth.

**Theoretical calculations of $\Delta G_H$.** The structural optimizations were carried out by adopting the generalized gradient approximation (GGA) with the Perdew–Burke–Ernzerhof (PBE) exchange-correlation functional, along with projector-augmented wave (PAW) potentials. The electronic wave functions were expanded in a plane wave basis set with a kinetic energy cutoff of 300 eV. For the Brillouin zone integration, $1 \times 5 \times 1$ Monkhorst–Pack $k$-point meshes were used. The energy convergence criterion for the electronic wavefunction was set at 10⁻⁵ eV. A vacuum distance of about 10 Å, both laterally between MoS₂ ribbons and vertically between layers, was chosen when constructing the supercell to minimize possible spurious interaction between ribbons due to the periodic boundary conditions.

The hydrogen adsorption energy is defined as:

$$\Delta E_H = E_{(MoS_2+H)} - E_{(MoS_2)} - {}^1\!/_2 E_{(H_2)} \qquad (1)$$

where $E_{(MoS_2+H)}$ is the energy of the MoS₂ system with a H atom adsorbed, $E_{(MoS_2)}$ is that of the MoS₂ system without adsorbed H, and $E_{(H_2)}$ is that of gas phase H₂ molecule.

The hydrogen adsorption free energy $\Delta G_H$ was defined as:

$$\Delta G_H = \Delta E_H + \Delta E_{ZPE} - T\Delta S_H \qquad (2)$$

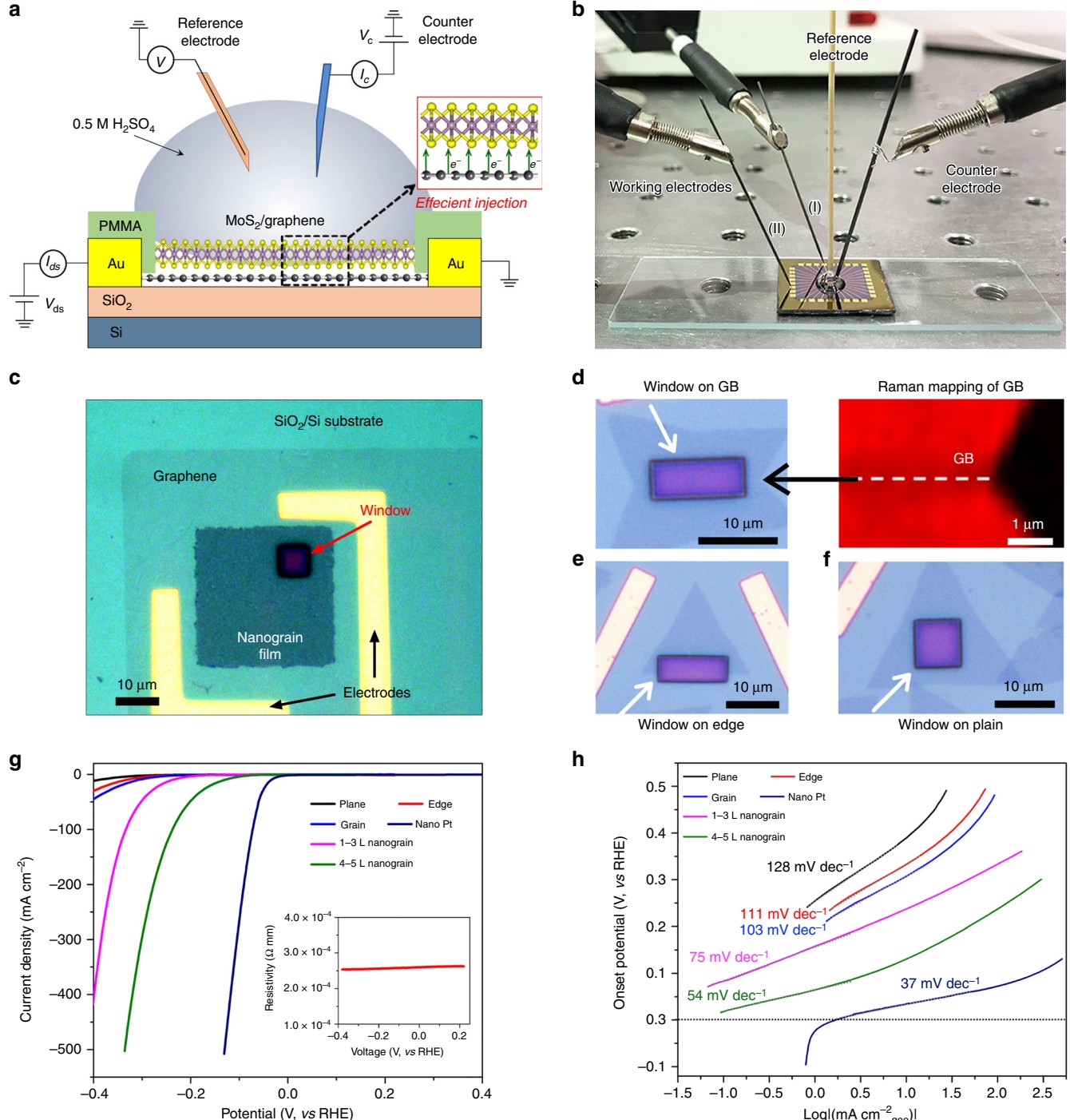

**Fig. 5 Micro-electrochemical cell-based HER activity of a MoS₂ nanograin film. a** Schematic of the micro-electrochemical cell for HER measurements, where graphene serves as a vertical electron injector. **b** Photograph of the micro-electrochemical cell. **c** Optical images of the MoS₂ nanograin device, consisting of a PMMA reaction window, a MoS₂ nanograin film, a graphene supporting layer, and a SiO₂/Si substrate from top to bottom. **d–f** Optical images of the MoS₂ devices with a single GB (**d**), a single edge (**e**), and basal plane (**f**), respectively. In these devices, the HER process occurs at the exposed widows on the PMMA passivation film as indicated by the white arrow. **g, h** Polarization curves of the current density (**g**) and the corresponding Tafel plots (**h**) of the MoS₂ devices. The window size is ~80 μm² for each device.

where $\Delta E_H$ is the hydrogen adsorption energy, $\Delta E_{ZPE}$ is the zero-point energy difference, $T$ is the temperature, and $\Delta S_H$ is the entropy difference for hydrogen between the adsorbed state and the gas phase. The entropy of hydrogen adsorption is calculated as $\Delta S_H = 1/2 S_{H_2}$, where $S_{H_2}$ is the entropy of hydrogen molecule in the gas phase at standard conditions. We have calculated the vibrational frequencies of H adsorbed on different MoS₂ systems, using finite differences to determine the Hessian matrix. According to the Sabatier principle, a good catalyst

should bind a key intermediate in a manner that it is strong enough to allow the reagent H atoms to stay on the MoS₂ system for reaction but weak enough to enable the release of any produced H₂ molecules.

**Phase-field simulations: simulation of the drive mechanism: MoS₂ pushing Au QDs long the growth direction**. The equilibrium status of the Au QDs at atomic

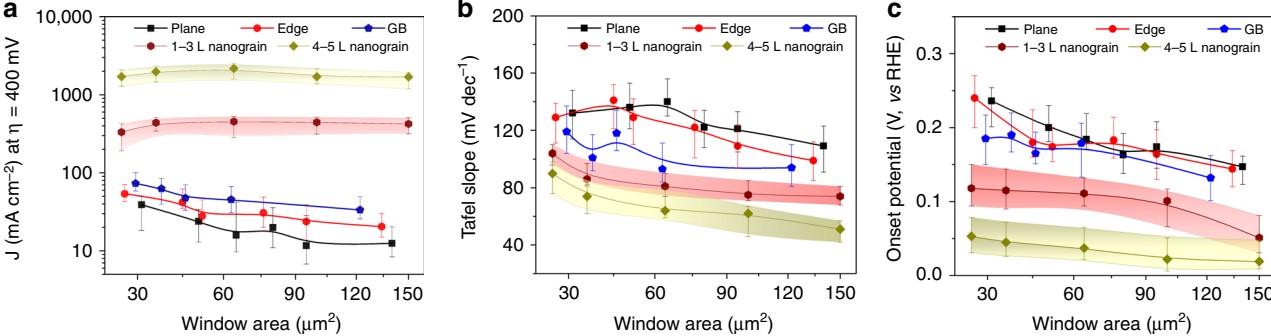

**Fig. 6 Comparison of the HER performances in a size-controlled micro-electrochemical cell. a–c** Current density (**a**), Tafel plots (**b**), and onset potentials (**c**) for the $MoS_2$ devices with serial window sizes from 25–150 $\mu m^2$. The nanograin films show higher current density, lower Tafel slope, and onset potential compared to the basal plane, single edge, and single GB.

steps dictates they can be pushed by the $MoS_2$ growth front or not, which is crucial to dynamic evolution process. We undertook a case study of the equilibrium status of a Au QD at the edge of a $MoS_2$ grain.

We adopt a phase-field model[63] to simulate how the wetting process influences the equilibrium shape profile. The free energy function is given as,

$$F(\rho) = \int_V [\frac{\varepsilon}{2}\rho^2(\rho-1)^2 + \frac{\kappa}{2}(\nabla\rho)^2 + \lambda\xi(\frac{\rho^3}{3} - \frac{\rho^2}{2})]dV \quad (3)$$

where $\rho$ is the phase field for the density of the liquid, $V$ is the space volume, $C = 10$ and $\kappa = 1$ are constants of bulk energy and surface tension, respectively. $\rho = 1$ denotes the liquid phase and $\rho = 0$ denotes the vapor phase. The term $\lambda\xi(\frac{\rho^3}{3} - \frac{\rho^2}{2})$ provides the driving force to the Au particle evolution with a coupling constant $\lambda = 180$, the dimensionless supersaturation $\xi$ is set as $(v_0 - v_t)/v_{vapor}$, where $v_0$ is the initial volume of the particle, $v_t$ is the real time volume of the particle, and $v_{vapor}$ is the volume of vapor. The evolution equation in the Allen–Cahn form is,

$$\frac{d\rho}{dt} = \kappa\Delta\rho - C\rho(\rho-1)(2\rho-1) + \lambda\xi(\rho-\rho^2) \quad (4)$$

This evolution equation differs from that of Borcia et al.'s work[63], which considers full fluid dynamics by including extra terms from the Navier–Stokes equation. In our study, we focus on the equilibrium and do not need to consider the exact evolution kinetics, thus leading to a simpler form of the evolution equation.

We control the contact angle between the Au QD and the substrate by fixing $\rho = \rho_S$ at the substrate. Then, the contact angle can be analytically obtained via the equation $\cos\theta = -1 + 6\rho_S^2 - 4\rho_S^3$. We set $\rho_S = 0.5$ for the $MoS_2$ substrate as corresponding to a contact angle between the Au and the substrate of 90°. Likewise, we set $\rho_S = 0.21$ for the $MoS_2$ substrate as corresponding to a contact angle between the Au and the substrate of 138°.

We solve the evolution equation in Matlab by discretization with a time step of 0.01 and let the simulation run for a long enough time until the system reaches equilibrium. The particle has a diameter of ~11 nm if resting on the $MoS_2$ sheet.

The result shows that this Au QD will stay on the lower part of the $MoS_2$ atomic step, but keeps in contact with the step edge. This suggests that if in the liquid phase, the Au QD would be pushed forward by the growth front of the second $MoS_2$ layer during the growth.

**Phase-field simulations: simulation of the zipper effect suturing the neighbor nanograins.** If the Au QDs can be pushed by the growing $MoS_2$ grains, they are likely to coalesce when two growth fronts meet, resulting in a complicated evolution. To obtain insights into this process, we perform another phase-field simulation. The phase-field model for the grain evolution and the set of parameters is almost identical to that reported in the literature for a similar system[64]. We ignore the evolution of the supersaturation and keep it at a constant of 0.1 everywhere. A detailed formulation is omitted here for brevity and can be found in other work[65,66]. In addition to the grains, we pay attention to the evolutions of the Au QDs. These QDs are modeled as hemispherical balls sliding on the substrate. For simplicity, we ignore the effect of the Au QDs on the growth of $MoS_2$ grains. Then these QDs move on the substrate at a speed of $v = 50 \tan^{-1}(t)/\pi$, where $t$ is the dimensionless time of evolution. When two QDs meet, they merge and form a larger QD. The volume of the new QD is the sum of the volumes of the previous two QDs and its center is at the midpoint between the centers of the previous two QDs. We then solve the evolution equation in Matlab by discretization with a time step of 0.01. Initially Au QDs are randomly placed on the substrate and there is in the vicinity of these QDs, a circular nucleus of $MoS_2$ with a radius of $r_0$ and a randomly assigned grain orientation.

To make the simulation consistent with the experiment, we extract some parameters from the experimental data. The experiment shows that the QDs are typically 4–5 nm. It also shows that when observed on top of the $MoS_2$ sheet, the Au QDs are larger and less dense by around 2.5 times. This suggests by pushing Au

QDs on top of $MoS_2$ sheet, the final number of QDs will be 2/5 of the initial QDs. A subtle issue involved in the simulation is the ratio between $n_{Au}$ (initial number of Au QDs) and $n_n$ (the number of nuclei). To address this issue, we take a simulation case study to find a reasonable ratio. By fixing $n_n = 5$, we obtain that when $n_{Au} = 9.2$, the final number of QDs is 2/5 of the initial QD number, consistent with the experiments. We therefore choose $n_{Au} = 10$ in further simulations.

**Device fabrication procedure.** First, large scale and high-quality single-layer graphene films were grown on Cu foils by CVD, and then transferred onto the prepatterned chips through a standard PMMA-assisted transfer method. Second, electron beam lithography (EBL) and $O_2$ plasma were employed to pattern the graphene film into isolated strips with desired size, shape, and location. Third, as-grown $MoS_2$ films were transferred to the patterned graphene strips, also using a PMMA-assisted transfer method. For the nanograin films, an additional step carefully positioning an EBL-defined PMMA template was needed. Fourth, an annealing process at 200 °C under high vacuum condition ($1 \times 10^{-5}$ Torr) was conducted to remove trapped residual molecules between the graphene and $MoS_2$ films to optimize their interfaces. Fifth, EBL followed by thermal evaporation was employed to fabricate the electrodes (Cr/Au, 2 nm/60 nm) on graphene to connect the device with the Au contact pads. Finally, the entire device as fabricated on the chip was passivated by a 500 nm PMMA film, followed by EBL to remove the PMMA above the region of interest in the $MoS_2$ film, and expose this section and this section only of the active catalyst to the electrolyte in HER test. Any electrochemical activity only occurs within the exposed window on the nanosheet, while the rest of regions, including the electrodes and the contact pads, are fully passivated by the PMMA to ensure a full electrochemical inertness.

A mirror GB in CVD-grown $MoS_2$ is chosen as representative of single GBs, due to the ease with which it can be distinguished in optical images. It also consists mainly of 8|4|4 or defect 8|4|4 rings[3,11] and has therefore a similar atomic structure to many of the complex boundaries observed in the nanograin films. The length–width ratio of the exposed window for the single-GB device or for the edge device (exposing the edge of an otherwise single-crystalline patch of CVD-grown $MoS_2$) is fixed at a ratio of about 2:1, and the length of the window is comparable to the length of the edge or GB in experiments.

**Four-electrode micro-electrochemical measurements.** A micro-electrochemical cell with four electrodes (the circuit diagram for this cell is introduced in Supplementary Fig. 20) was developed in our experiment. Among the four electrodes, two serve as counter and reference electrode, using pencil graphite and a Ag/AgCl micro reference electrode (Harvard Apparatus), respectively. The remaining two electrodes were connected to the graphene supporting layer to monitor the conductance signal of graphene and the electrocatalytic signal of $MoS_2$ during HER. The device fabrication procedure is shown in Supplementary Figs. 21 and 22, and Supplementary Notes 2 and 3. In all experiments, only the exposed region of the $MoS_2$ nanosheets contributes to the recorded HER performance.

The typical four-electrode micro-electrochemical measurements were conducted in a 0.5 M $H_2SO_4$ electrolyte solution. The scan rate is 5 mV step$^{-1}$. Both electrocatalytic current ($I_c$) and conductance current ($I_{ds}$) are simultaneously detected. The leakage current in $I_c$ is about $10^{-10}$ A (the electrochemical signal without any exposed window, i.e., from a region with PMMA passivation). The electrochemical current density is calculated by normalizing the current to the area of the exposed window on the $MoS_2$. In 0.5 M $H_2SO_4$ solution, we measured:

$$E_{RHE} = E_{Ag/AgCl} + 0.219 \, V \quad (5)$$

**Standard macro-electrochemical measurements.** The macro-electrochemical measurements were conducted on a glassy carbon electrode (3 mm in diameter). Using a similar PMMA-assisted transfer method as described above, we transferred CVD-grown graphene and $MoS_2$ nanograin films layer-by-layer on a glassy carbon

electrode, and the sample area exceeding to glassy carbon electrode was scraped off before measuments. A standard three electrode system was used, and a Pt plate and Ag/AgCl rod served as counter and reference electrode, respectively. The measurement was performed on a biological electrochemical station in $H_2$-saturated 0.5 M $H_2SO_4$ solution. Linear sweep voltammetry (LSV) was conducted at a scan rate of 5 mV s$^{-1}$. The onset potential is defined as the beginning potential of Tafel linear region. The stability test was carried out by taking continuous potential cycling in the potential window of −0.181 to ~0.219 V (versus RHE) at a scan rate of 100 mV s$^{-1}$. The presented LSV curves in macro-electrochemical cell were iR-corrected by subtracting the ohmic resistance loss (about 9 Ω), the value of which is obtained from electrochemical impedance spectroscopy measurement (Supplementary Fig. 32). To evaluate the electrochemically active surface area of the catalysts, the Cu underpotential deposition method[67–70] was applied, as shown in Supplementary Fig. 30 and Supplementary Table 3.

To test the faradaic efficiency, the $H_2$ products were analyzed with an online gas chromatography (GC) setup (Agilent 7890B) equipped with a thermal conductivity detector. Argon (≥99.999%) was used as the carrier gas. The Faradaic efficiency (FE) was calculated by comparing the measured amount of $H_2$ generated by cathodal electrolysis with the calculated amount of $H_2$ (assuming an FE of 100%), and the equation is given by:

$$FE(\%) = \frac{96485 \times 2 \times \text{mol } H_2(GC) \times 100}{Q} \tag{6}$$

where the moles of $H_2$ is measured by GC (the calibration is needed in advance through injecting high-purity $H_2$ to GC, and the $H_2$ volume is linearly dependent on GC peak area), Q is obtained from the electrochemical measurements.

**Material characterization**. The microstructures and morphologies based on $MoS_2$ were characterized by optical microscopy, SEM (FEI 4200), and Raman spectroscopy (Renishaw inVia). Raman spectroscopy was performed with a 514.5 nm laser (with spot size about 1 μm in diameter) at room temperature. The micro-electrochemical measurements were performed using two source meters (Keithley 2400 and 2450). STEM measurements were performed at the SuperSTEM Laboratory, Daresbury, UK, on a Nion UltraSTEM100 aberration-corrected dedicated STEM. The Nion UltraSTEM has a cold field emission gun with a native energy spread of 0.35 eV and was operated at 60 kV acceleration voltage. The beam was set up to a convergence semiangle of 30 mrad with an estimated beam current of ~100 pA. Under these operating conditions, the estimated probe size is ~1.1 Å, providing the perfect tool for atom-by-atom chemical analysis. HAADF imaging was employed to produce atomically resolved images whose intensity is approximately proportional to the square of the average atomic number Z of the material under investigation. This chemically sensitive "Z-contrast" mode is ideally suited to directly identify the nature of individual atoms. All the HAADF images presented here were filtered for visual clarity using a wiener filter with coefficient sigma ranging from 2–5 and 50 iterations in MEM (Maximum Entropy Methods) with fourth order Gaussian filter, as implemented in the STEM-CELL software[71,72]. Geometrical Phase Analysis (GPA) was carried out using a cosine mask of suitable size and 1 binning base. HRTEM and low-resolution annular dark field (ADF)-STEM images were obtained on a FEI Tecnai F20 field emission gun microscope with a 0.19 nm point-to-point resolution at 200 kV, equipped with an embedded Quantum Gatan Image Filter for EELS analyses. Some STEM characterizations were carried out on a JEOL ARM-200F (S)TEM equipped with CEOS CESCOR aberration corrector, operated at an accelerating voltage of 80 kV. The convergence semiangle and acquisition semiangle were 28–33 and 68–280 mrad for the ADF imaging. The dwell time per pixel was set to 12–20.

## Data availability

The data that support the findings of this study are available from the corresponding author upon reasonable request.

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

## Acknowledgements

Z.L. gratefully acknowledges funding supports from Ministry of Education (MOE) under AcRF Tier 1 (M4011782.070 RG4/17 and M4011993.070 RG7/18), AcRF Tier 2 (2015-T2-2-007, 2016-T2-1-131, 2016-T2-2-153, and 2017-T2-2-136), AcRF Tier 3 (2018-T3-1-002), National Research Foundation – Competitive Research Program (NRF-CRP21-2018-0092), and A*Star QTE programme. P.T., S.M.S., J.R.M., and J.A. acknowledge funding from Generalitat de Catalunya 2017 SGR 327 and 1246 and the Spanish MINECO coordinated projects between IREC and ICN2 VALPEC (ENE2017-85087-C2-C3). ICN2 acknowledges support from the Severo Ochoa Program (MINECO, Grant SEV-2017-0706). IREC and ICN2 are funded by the CERCA Programme/Generalitat de Catalunya. Part of the present work has been performed in the framework of Universitat Autònoma de Barcelona Materials Science PhD program. S.M.S. acknowledges funding from 'Programa Internacional de Becas "la Caixa"-Severo Ochoa'. J.R.M. recognizes also its affiliation to University of Barcelona. Part of the electron microscopy aspects of this work were supported by the EPSRC (UK), as the SuperSTEM Laboratory is the EPSRC National Research Facility for Advanced Electron Microscopy. Q.J.W. acknowledges the supports from MOE, Singapore grant (MOE2016-T2-2-159, MOE2016-T2-1-128, and MOE Tier 1 RG164/15) and National Research Foundation, Competitive Research Program (NRF-CRP18-2017-02) and NSFC (61704082) as well as Natural Science Foundation of Jiangsu Province (BK20170851). X.W. and B.K.T. gratefully acknowledge funding support from MOE, Singapore (grant no. MOE2015-T2-2-043). H.Z. acknowledges the supports from MOE under AcRF Tier 2 (MOE2015-T2-2-057; MOE2016-T2-2-103; and MOE2017-T2-1-162), AcRF Tier 1 (2016-T1-002-051; 2017-T1-001-150; and 2017-T1-002-119), Agency for Science, Technology and Research (A*STAR) under its AME IRG (Project No. A1783c0009), and NTU under Start-Up Grant (M4081296.070.500000) in Singapore. The authors would like to acknowledge the Facility for Analysis, Characterization, Testing, and Simulation, Nanyang Technological University, Singapore, for their electron microscopy and X-ray facilities. H.Z. also thanks the support from ITC via Hong Kong Branch of National Precious Metals Material Engineering Research Center, and the Start-Up Grant from City University of Hong Kong. Theory and simulations work at Rice university (Z.H., L.W., and B.I.Y.) was supported by the Office of Naval Research grant N00014-18-1-2182.

## Author contributions

Z.L. and Y.H. conceived and initiated the project. Z.L., Q.W., J.A. and M.W. supervised the project and led the collaboration efforts. Y.H. designed the experiments, grew nanograin film, fabricated the electrochemical devices, and performed micro-electrochemical HER measurement. P.T., C.Z. (Dr. Chao Zhu), S.M.S., J.A., J.M. and Q.M.R. performed the TEM and STEM measurements. Z.Hu did the phase-field simulation of the growth process. Y.H., Q.H. and C.Z. (Mr. Chao Zhu) made the micro-electrochemical measurement setup. L.W. (Luqing Wang) and B.Y. performed the first-principles calculations of the catalytic activity. Q.Z. did the SEM measurement. G.G. and A.T. conducted the cross-sectional TEM measurement. Z. Huang and Q.H. performed gas chromatography measurement. P.G., X.W. (Xuewen Wang), W.F., C.G., X.W. (Xingli Wang), B.A., Y.Z., L.W. (Liang Wang), and B.T. assisted the growth of nanograin film. J.X. and A.Z. did the Raman measurements. Y.H., P.T., Z.Hu, Q.W., H.Z., J.A. and Z.L. wrote the paper. All authors discussed the results and commented on the manuscript.

## Competing interests

The authors declare no competing interests.
