## [Peer Review File · Nature Communications]

Reviewers' comments:

Reviewer #1 (Remarks to the Author):

In this manuscript, the author reports a new way to connect HER performance with the concentration and structure of GB for improved HER performance of MoS₂. The reported procedure, which uses Au QDs to induce growth and control the concentration of GB, is an interesting approach. According to the previous reviewer's comments, the revised manuscript is quite improved by the supplemental data, such as the bulk electrochemical data, etc. However, some key points in this work still need to be further addressed. Thus, I would suggest the following revisions be resolved prior to acceptance for publication.

1. An important control experiment missing in this paper is testing for the presence of Au single-atoms after removing the Au QDs. Single-atom catalysts decorated onto MoS₂ have been proven to enhance catalytic performance (Energy. Environ. Sci., 2015, 8, 1594; Applied Catalysis B: Environmental, 2019, 251, 87). The presence of Au single-atoms may have a similar effect on the work discussed herein. Several bright spots are visible in the left figure within Fig. 2B, which may correspond to the presence of Au single-atoms in the catalyst.

2. Since the grain size and density are seldom discussed in the previous works, the author should explain how the data in Fig. 1A was summarized and calculated.

3. The author's conclusion of how an increase in GB concentration would result in better catalytic performance is very rational based on the fact that more GB will induce greater exposure to edge sites. However, a more in-depth discussion of the concepts that lead to this conclusion is necessary. For example, how to distinguish differences between GB, GB induced edge sites and edge sites originated not from GB?

4. In terms of the electrochemical performance, the MoS₂ nanograin films exhibit low onset potentials. The author should discuss how the onset potential was defined and measured for these experiments. Moreover, all LSV data in the paper is presented after IR correction – using different values for %IR correction will produce inconsistent results. Thus, IR correction parameters should be presented in the paper as well. The performance comparison provided in Table S3 is meaningless if onset potential and IR correction parameters used in the papers referenced for comparison do not match the parameters used in the manuscript.

5. The author should provide a clearer explanation of the difference between each type of GB discussed and how are they defined, such as 5/7 GB, 8/4/4 GB, etc.

Reply to Reviewer #1

In this manuscript, the author reports a new way to connect HER performance with the concentration and structure of GB for improved HER performance of MoS₂. The reported procedure, which uses Au QDs to induce growth and control the concentration of GB, is an interesting approach. According to the previous reviewer's comments, the revised manuscript is quite improved by the supplemental data, such as the bulk electrochemical data, etc. However, some key points in this work still need to be further addressed. Thus, I would suggest the following revisions be resolved prior to acceptance for publication.

Response: We are very thankful for the Reviewer's positive comments on our work.

Comment 1. An important control experiment missing in this paper is testing for the presence of Au single-atoms after removing the Au QDs. Single-atom catalysts decorated onto MoS₂ have been proven to enhance catalytic performance (Energy. Environ. Sci., 2015, 8, 1594; Applied Catalysis B: Environmental, 2019, 251, 87). The presence of Au single-atoms may have a similar effect on the work discussed herein. Several bright spots are visible in the left figure within Fig. 2B, which may correspond to the presence of Au single-atoms in the catalyst.

Response: We thank the Reviewer for this invaluable comment. We fully agree with the Reviewer on "single-atom catalysts decorated onto MoS₂ shows enhanced catalytic performance", such as Pt, Co, Ni, and Cu in above papers. In our case, in order to clarify the influence of Au single-atoms on the electrocatalytic performance of MoS₂ nanograin film, we have conducted two series of experiments: (1) use STEM/XPS characterizations to examine the presence of Au single-atoms after the removal of the Au QDs; (2) investigate the HER activity of Au in various nanostructures and compare it to our MoS₂ nanograin film;

(1) Examine Au single-atoms by STEM/XPS.

Firstly, in order to clarify the nanostructure of residual Au (individual atoms or small clusters) after removing the Au QDs, we used STEM to carefully exam the MoS₂ nanograin film. **Figure R1** shows that only few residual Au single atoms (white bright spots) (**a-c**) and a very small number of nanoparticles (**d**) can be found in the film. It is shown that Au single atoms are mainly

located at the edge instead of the crystal structure (basal plane) of MoS₂. This is *different from the aforementioned single-atom catalysts on MoS₂* where the metal atoms usually occupy the positions of the Mo atoms on the basal plane of the nanosheets (Energy. Environ. Sci., 2015, 8, 1594). This may be due to a much lower diffusion of Au than that of the fast gaseous-kinetic growth processes during the MoS₂ growth process. It is worth mentioning that *doping Au atom into MoS₂ structure appears to be difficult in a CVD process*. Many other works have reported the use of Au foils or thin-films to grow MoS₂, WS₂, or WSe film,¹⁻⁵ however no presence of Au single-atom was found in these TMD materials.

Figure R1 (Figure S6). Atomically resolved HAADF STEM images of the MoS₂ nanograin film after removal of Au. (a-c) Au atoms (indicated by brown arrows) at the edge of grains (a-b) or hollow spheres (c). The small white bright spots correspond to Au atom. The MoS₂ hollow sphere is derived from the removal of Au nanoparticles. From the STEM images, only a very small amount of Au atoms can be observed, suggesting that most of Au has been etched. (d) Au nanoparticles, which are fully coated by MoS₂. It is difficult to remove Au due to the protection of MoS₂. More often, we observed hollow spheres which indicate that the Au nanoparticles inside has been successfully etched.

Secondly, in order to evaluate the overall content of residual Au, we employed XPS to investigate the MoS₂ nanograin film before and after the removal of Au, as shown in **Figure R2**. After removing Au by KI/I₂ etcher in our experiment condition, only very a small amount of Au (about 0.86% atomic ratio) remains on the MoS₂ film, and *the amount of Au single-atom should be much less due to the existence of larger residual Au nanoparticle*. The XPS data are listed in

Figure S7 in Supplementary Information.

Figure R2 (Figure S7). XPS of MoS₂ nanograin film before and after removal of Au. (a-c) XPS spectra of Au4f, Mo3d, and S2p before and after removing Au by KI/I₂ etcher. (d) Atomic ratio in MoS₂ nanograin film before and after removal of Au. A 0.86% of Au atomic ratio in film suggests KI/I₂ etcher can effectively remove the Au nanoparticles in MoS₂ nanograin film.

(2) Investigate the HER activity of Au

We have summarized previous reports on the HER performance of Au, among which Au nanoparticles and small clusters show moderate HER performance (Onset potentials of -120 ~ -250 mV, Tafel slopes of 73 ~ 121 mV/dec), yet still much lower than our MoS₂ nanograin film (Onset potential of -44 mV, Tafel slope of 55 mV dec⁻¹), as shown in **Table R1**.

Types	Onset potentials (V, vs RHE)	Tafel slops (mV dec ⁻¹)	References
Au thin film (80 nm)	-0.25	121	6,7
Au thin film (100 nm)	-0.22	88	8
Au (111) substrate	-0.21	73.8	9
Au foil	-0.2 ~ -0.22	117	1-3

Au nanocluster	- 0.2	73.9	¹⁰
Au nanocluster	- 0.12	96	¹¹
Au nanoparticle	- 0.19 ~ - 0.21	75-80	Our result
Au thin-film electrode (60 nm)	- 0.195	109	Our result
MoS ₂ nanograin film	- 0.044	55	Our result

Table R1. Summary of HER activity of Au from previously reports and our work.

In order to measure the HER performance of Au used in our CVD growth, we deposited the same Au onto carbon cloth, then followed the same growth condition of MoS₂ nanograin film but without Mo and S sources, and finally measured its HER performance, as shown in **Figure R3**. Therefore, we can make a clear comparison of the HER performance of Au with and without MoS₂. It is clearly seen that the Au nanoparticle on carbon cloth delivers a moderate HER performance (onset potential of - 195 mV, Tafel slope of 77 mV dec⁻¹) agreeing with previous reports (**Table R1**), which is much lower than MoS₂ nanograin film (onset potential of - 44 mV, Tafel slope of 55 mV dec⁻¹). The information of Au HER-activity is listed in **Figure S33** in Supplementary Information.

Figure R3 (Figure 33). HER activity of Au nanoparticles in our experiment. (a-b) SEM of Au nanoparticle on carbon cloth. A 0.3 nm of Au was deposited on carbon cloth and then followed the growth condition of MoS₂ without Mo and S sources. (c-d) The polarization curves (c) and the corresponding Tafel plots (d) of Au nanoparticles and MoS₂ nanograin as comparison.

To further support our conclusion, we further examined the activity of the residual Au by completely removing MoS₂ using O₂ plasma (20 W for 10 ~ 20 min etching time) and conducted HER measurement again on the remaining Au. **Figure R4** shows that the remaining Au delivers a negligible HER performance compared to original MoS₂ nanograin. This information is added in **Figure S34** in Supplementary Information.

Figure R4 (Figure S34). HER measurement of the film after removing MoS₂ by O₂ plasma on glassy carbon electrode.

In order to give a clear description, we add the sentences “*It is worth mentioning that Au single-atom exists in our MoS₂ nanograin film and may contribute to the overall HER performance. However, based on the structure and extremely low content of Au single-atom as well as its low HER-activity, we conclude that the residual Au is not the main contributor of the good HER activity in MoS₂ nanograin film.*” in 1st paragraph on Page 11.

Comment 2. Since the grain size and density are seldom discussed in the previous works, the author should explain how the data in Fig. 1A was summarized and calculated.

Response: We appreciate the Reviewer’s careful suggestion. We collected the data of grain size and density from two types of TMD samples: continuous film and insolated single crystal, as shown in

Figure R5a and b respectively. In general, continuous TMD films are usually obtained by the methods of CVD, MOCVD, “top-down” syntheses of sulfurization/selenization of metal and metal oxide thin films, and thermal decomposition of thiosalts thin films. And insolated crystals are usually made by CVD and PVD methods.

In our collection, the data of grain sizes are obtained from optical, SEM, or TEM/STEM images. The data of grain density for the continuous film is calculated on grains per unit area ($\frac{1 \text{ cm}^2}{\text{grain area}}$), as shown in **Figure R5a**. However, as for the insolated single crystal, the number of their GBs can be considered 1 (**Figure R5b**) and is independent of grain size. In order to give a clear description, we added this collection information of the grain size and density in **Figure S1**.

Figure R5 (Figure S1). Schematic of grain size and density collected from TMD continuous film (a) and insolated single crystal (b).

Comment 3. The author’s conclusion of how an increase in GB concentration would result in better catalytic performance is very rational based on the fact that more GB will induce greater exposure to edge sites. However, a more in-depth discussion of the concepts that lead to this conclusion is necessary. For example, how to distinguish differences between GB, GB induced edge sites and edge sites originated not from GB?

Response: Thanks for the Reviewer’s insightful suggestion. Both GB and edge can be considered “line defects” in MoS₂.¹² Compared to edge with unsaturated S or Mo bonds, GB is comprised of Mo-S rings (i.e., 5|7 ring, 6|8 ring, 4|6 ring, 8|4|4 ring) with strong intrinsic strain. Because of their distinct

atom structures, GB and edge usually have different hydrogen adsorption sites, please see **Figure R6** for their varied structures and adsorption sites. As for GB induced edge that Reviewer mentioned, we think it would similarly be also one kind of Mo or S edges derived from the broken Mo-S rings that release the strain via external methods (transfer, annealing, fast cooling, etc).

We further compared the HER activities of most of edges, i.e., S edge with 100%, 75%, and 50% S-passivation and Mo edge with 100%, 50%, and 0% S-passivation, and GBs, i.e., 5|7 GB, 6|8 GB, 4|6 GB, 8|4|4 GB, Defected 8|4|4 GB, 12|4 GB, as shown in **Figure 3**. It is seen that GBs indeed show even better activity than the edges in general, which is consistent with our measurements on single MoS₂ GB and edge in electrochemical microcell (**Figure R7**), suggesting that GB is a promising candidate for highly efficient catalytic active sites.

Figure R6 (Figure S18). Schematic of hydrogen adsorption on various atomic structures in MoS₂.

(a)-(c) S edge with 100% (a), 75% (b), and 50% (c) S-passivation. (d)-(f) Mo edge with 100% (d), 50% (e), and 0% (f) S-passivation. (g) 5|7 GB. (h) 6|8 GB. (i) 4|6 GB. (j) 8|4|4 GB. (k) Defected 8|4|4 GB. (l) 12|4 GB. Green spheres represent Mo atoms, yellow spheres represent S atoms and blue spheres, H.

In order to elaborate the comparison between GB and edge, we have individually examined the HER activity of single GB and edge in electrochemical micro-cell system. The results show that GB possesses better HER activity than edge site, as shown in **Figure R7** (see also in Figures 5 and 6 in the main text for details). It is worth mentioning that although we cannot exclude the contribution from edge sites to the overall HER activities in our nanograin films, we believe that GBs is responsible for the excellent HER performance with its extremely high density and superior activity, which is verified by our TEM/STEM, theoretical calculations and micro-cell experiments.

Figure R7. HER performances of single-edge and single-GB in a micro-electrochemical cell. (a-b) Optical images of the MoS₂ devices with a single GB (a) and a single edge (e), respectively. In these devices, the HER process occurs at the exposed windows on the PMMA passivation film as indicated by the white arrow. (c-d), Polarization curves of the current density (c) and the corresponding Tafel plots (d) of the MoS₂ devices. The window size is about 80 μm² for each device.

Comment 4. In terms of the electrochemical performance, the MoS₂ nanograin films exhibit low onset potentials. The author should discuss how the onset potential was defined and measured for these

experiments. Moreover, all LSV data in the paper is presented after IR correction – using different values for %IR correction will produce inconsistent results. Thus, IR correction parameters should be presented in the paper as well. The performance comparison provided in Table S3 is meaningless if onset potential and IR correction parameters used in the papers referenced for comparison do not match the parameters used in the manuscript.

Response: We appreciate the Reviewer’s nice suggestions. Following the Reviewer’s suggestion, we have added a detailed description of the onset potential, and please see “*The onset potential is defined as the beginning potential of Tafel linear region*” in Experiment Section.

In our manuscript, only LSV data in conventional macro-electrochemical cell have used %iR correction, and the value of iR is obtained from EIS measurements on conventional glass-carbon electrode, as shown in **Figure R8**. We also added iR correction parameters in Experiment Section “*The measured results in macro-electrochemical cell have been iR-corrected by subtracting the ohmic resistance loss (about 9 Ω), the value of which is obtained from EIS measurement*” and in **Figure S32** in Supplementary Information. On the other hand, iR correction is not applied in the all the micro-cell measurements, due to the very small signal (nA) in micro-electrochemical cell.

Figure R8 (Figure S32). Electrochemical impedance spectroscopy (EIS) measurement of MoS₂ nanograin film (a) and (b). R is about 9 Ω , obtained from (b).

Finally, we have double check all of the References in Table S3, and find that some of them (*Nat. Mater.* **15**, 1003-1009 (2016); *J. Am. Chem. Soc.* **135**, 10274-10277 (2013); *Proc. Natl. Acad. Sci. U.S.A.* **110**, 19701-19706 (2013); *J. Am. Chem. Soc.* **138**, 7965-7972 (2016).) used the iR correction while some did not so. In order to avoid confusion, we delete this table in the revised manuscript.

Comment 5. The author should provide a clearer explanation of the difference between each type of GB discussed and how are they defined, such as 5/7 GB, 8/4/4 GB, etc.

Response: Thanks for the Reviewer's suggestion. 5|7 GB is formed from 5- and 7- membered rings, which is highlighted by the overlaid red pentagon and purple heptagon in **Figure R9a**, with a recurring periodic 5-7 ring motif. 6|8 GB is formed from 6- (red hexagon in **Figure R9b**) and 8- (purple octagon in **Figure R9b**) membered rings, with a recurring periodic 6-8 ring motif. 4|6 GB is formed from 4- (red tetragon in **Figure R9c**) and 6- (purple hexagon in **Figure R9c**) membered rings, with a recurring periodic 4-6 ring motif. 8|4|4 GB is formed from 8- (red octagon in **Figure R9d**) and two 4- (purple tetragons in **Figure R9d**) membered rings, with a recurring periodic 8-4-4 ring motif. Defected 8|4|4 GB is formed by removing two sulphur atoms which connect two 4-membered rings from 8|4|4 GB, as shown in **Figure R9e**. 12|4 GB is formed from 12- (red dodecagon in **Figure R9f**) and 4- (purple tetragon in **Figure R9f**) membered rings. We added **Figure S10** to give a clearer explanation of the difference between each type of GB.

Figure R9 (Figure S10). Schematic illustration of atomic structures of 5|7 GB, 6|8 GB, 4|6 GB, 8|4|4 GB, Defected 8|4|4 GB, and 12|4 GB in MoS₂.

References

- 1 Zhang, Y. *et al.* Chemical vapor deposition of monolayer WS₂ nanosheets on Au foils toward direct application in hydrogen evolution. *Nano Res.* **8**, 2881-2890 (2015).
- 2 Shi, J. *et al.* Controllable Growth and Transfer of Monolayer MoS₂ on Au Foils and Its Potential Application in Hydrogen Evolution Reaction. *ACS Nano* **8**, 10196-10204 (2014).
- 3 Zhang, Y. *et al.* Dendritic, Transferable, Strictly Monolayer MoS₂ Flakes Synthesized on SrTiO₃ Single Crystals for Efficient Electrocatalytic Applications. *ACS Nano* **8**, 8617-8624 (2014).
- 4 Gao, Y. *et al.* Large-area synthesis of high-quality and uniform monolayer WS₂ on reusable Au foils. *Nat. Commun.* **6**, 8569 (2015).
- 5 Gao, Y. *et al.* Ultrafast Growth of High-Quality Monolayer WSe₂ on Au. *Adv. Mater.* **29**, 1700990 (2017).
- 6 Li, H. *et al.* Activating and optimizing MoS₂ basal planes for hydrogen evolution through the formation of strained sulphur vacancies. *Nat. Mater.* **15**, 48-53 (2016).
- 7 Li, H. *et al.* Kinetic Study of Hydrogen Evolution Reaction over Strained MoS₂ with Sulfur Vacancies Using Scanning Electrochemical Microscopy. *J. Am. Chem. Soc.* **138**, 5123-5129 (2016).
- 8 Shin, S., Jin, Z., Kwon, D. H., Bose, R. & Min, Y.-S. High Turnover Frequency of Hydrogen Evolution Reaction on Amorphous MoS₂ Thin Film Directly Grown by Atomic Layer Deposition. *Langmuir* **31**, 1196-1202 (2015).
- 9 Jaramillo, T. F. *et al.* Identification of active edge sites for electrochemical H₂ evolution from MoS₂ nanocatalysts. *Science* **317**, 100-102 (2007).
- 10 Zhao, S. *et al.* Atomically Precise Gold Nanoclusters Accelerate Hydrogen Evolution over MoS₂ Nanosheets: The Dual Interfacial Effect. *Small* **13**, 1701519 (2017).
- 11 Zhang, J. *et al.* Molybdenum disulfide and Au ultrasmall nanohybrids as highly active electrocatalysts for hydrogen evolution reaction. *J. Mater. Chem. A* **5**, 4122-4128 (2017).
- 12 Zhou, W. *et al.* Intrinsic Structural Defects in Monolayer Molybdenum Disulfide. *Nano Lett.* **13**, 2615-2622 (2013).

Finally, we would like to thank the Reviewer again for careful review and very helpful suggestions such Au single-atom, difference between GB and edge, iR-correction, etc that indeed help us to improve the depth and quality of our manuscript greatly.

REVIEWERS' COMMENTS:

Reviewer #1 (Remarks to the Author):

The authors have answered all questions the reviewer has raised in detail. Thus, the acceptance of the publication of the manuscript is recommended.

Reply to Reviewer #1

The authors have answered all questions the reviewer has raised in detail. Thus, the acceptance of the publication of the manuscript is recommended.

Response: We highly appreciate the Reviewer's help and time for reviewing our work, and his/hers insightful suggestions greatly improve the depth and quality of our manuscript.